# Pneumatic Noise Study of Multi-Stage Sleeve Control Valve

**Jianbo Jia [1], Yan Shi [1,*], Xianyu Meng [1,*], Bo Zhang [2] and Dameng Li [2]**

[1] College of Mechanical and Electrical Engineering, Changchun University of Science and Technology, Changchun 130022, China; asher0202@126.com

[2] He Harbin Power Plant Valve Company Limited, Harbin 150000, China; zbqwj1983@sina.com (B.Z.); 15804627113@126.com (D.L.)

* Correspondence: shiyan@cust.edu.cn (Y.S.); mengxianyu@cust.edu.cn (X.M.)

**Abstract:** This study considers the practical issue of severe noise observed in a multi-stage sleeve control valve within an engineering project. Employing computational fluid dynamics (CFD) methodology, we initially performed numerical simulations to analyze the steady-state flow field within the control valve. Subsequently, we identified the underlying factors contributing to the noise issue within the valve. To assess the aerodynamic noise of the control valve, we applied the FW-H acoustic analogy theory and determined the intensity and distribution characteristics of the aerodynamic noise. Finally, we validated the numerical simulation results of the aerodynamic noise against theoretical calculations. Our findings indicate that the steam medium experiences high-speed flow due to disturbances caused by various components within the valve, resulting in significant turbulence intensity. This intense turbulence leads to pressure fluctuations in the steam, serving as the main catalyst for noise generation. The aerodynamic noise of the control valve exhibits a roughly symmetrical distribution along the pipe–valve system, with noticeable increases in noise levels upstream and downstream of the valve compared to other regions. The distribution cloud map obtained from the numerical simulations serves as a valuable reference for analyzing the locations where aerodynamic noise is generated. Comparing the numerical simulation results with the theoretical calculations at the noise monitoring points, we found that the noise error of the monitoring points was less than 5%, which proves the effectiveness of the numerical simulation method. These results provide essential data support for the acoustic detection of aerodynamic noise in control valves, carrying significant practical implications for engineering applications.

**Keywords:** control valve; flow field; computational fluid dynamics (CFD); aerodynamic noise; sound field

## 1. Introduction

Multi-stage sleeve control valves play a critical role as key control components in power plants and petrochemical systems. They are frequently employed in the transportation of high-temperature, high-pressure, and high-pressure differential media during industrial operations [1]. The design concept behind multi-stage sleeve control valves involves subjecting the medium to a series of pressure reductions as it flows through the valve, with each stage of the throttle sleeve contributing to a drop in pressure. The multi-stage throttling device, serving as a core component, effectively reduces excessive flow rates. By controlling pressure fluctuations within permissible ranges, the erosion of valve components caused by high-velocity fluid media is mitigated. Consequently, the service life of the valves is prolonged, ensuring the reliable operation of equipment and systems [1,2].

In engineering projects, the noise in control valve systems is commonly classified into three main categories based on their underlying mechanisms: mechanical vibration noise, hydrodynamic noise, and aerodynamic noise [3,4]. Specifically, aerodynamic noise is the term used to describe the dynamic noise generated by gas flows. This study focuses on the

complex structure of multi-stage sleeve control valves. When steam media passes through the throttling components, the rapid pressure changes result in intense turbulence within the valve, generating significant aerodynamic noise that can adversely affect the normal operation of the control valve system. Prolonged exposure to high levels of noise can cause distractions and emotional distress among equipment operators, and in severe cases, it can impact their physical and mental well-being [5]. Furthermore, excessive noise can mask potential equipment malfunctions, thereby increasing the risk of accidents.

The issue of valve noise has been extensively investigated by numerous scholars. From the perspective of the sound wave generation mechanism, Liu et al. [6] used CFD software 2014 coupled with acoustic software2014 to study the aerodynamic noise generated by gas flowing through the valve. Wu et al. [7] carried out a CFD numerical simulation of the flow field of the turbine bypass control valve and obtained the flow field inducements of noise and vibration. Liao et al. [8] employed a fluid–structure-acoustic coupling numerical simulation method to elucidate the mechanism of aerodynamic noise generation in eccentric rotating valves. Nord [9] analyzed the causes of the noise generated by the control valve and considered more cost-effective ways to reduce the noise. Using numerical simulations and analyses of multi-stage sleeve control valves at different openings, Sun et al. [10] identified the causes of aerodynamic noise in valves. In references [11,12], the pneumatic noise characteristics of the control valve were analyzed by combining experiment and numerical simulation. Fan et al. [13] described the generating mechanism of pneumatic noise of regulating valves and gave suggestions on noise suppression and elimination. Shi et al. [14] carried out numerical simulation research on valve noise of a Marine three-way regulating valve and obtained noise sound pressure spectrum characteristics and the sound directivity law. Sun et al. [15] obtained the internal sound field of the control valve by calculating the internal turbulent flow field and then solved the external sound field of the valve by using the acoustic vibration model. Xu et al. [16] analyzed the mechanisms and influencing factors of valve noise and proposed optimization designs for noise reduction. Li et al. [17] theoretically calculated and experimentally verified the aerodynamic noise generated by gas flow through valves. In accordance with the methods described in the international standard IEC 60534-8-3, Luca Fenini et al. [18] predicted the intensity of aerodynamic noise through orifice plates using internal sound power level and external sound pressure level (SPL) as indicators. Li et al. [19] introduced a noise calculation method for a multi-stage sleeve regulating valve and combined it with the standard IEC60534-8-3 to predict the pneumatic noise of the valve. Wei et al. [20] mainly studied the characteristics of flow-induced noise of pressure-reducing valves based on the computational fluid dynamics method, and put forward suggestions for valve noise control methods.

This study primarily employs computational fluid dynamics (CFD) methods to validate the causes and distribution of aerodynamic noise in control valves. The numerical simulation results are compared and analyzed against theoretical findings. The intensity of aerodynamic noise generated by the valve orifice plates is predicted. While it may not be possible to eliminate aerodynamic noise, reducing the noise level can be achieved by adjusting the structural design of the throttling elements. This research holds significance for engineering practice.

## 2. Valve Structure and Working Principle

This study presents the design and configuration of a control valve used for flow regulation, with a nominal diameter of DN125, Class900. Figure 1 illustrates the three-dimensional representation of the control valve.

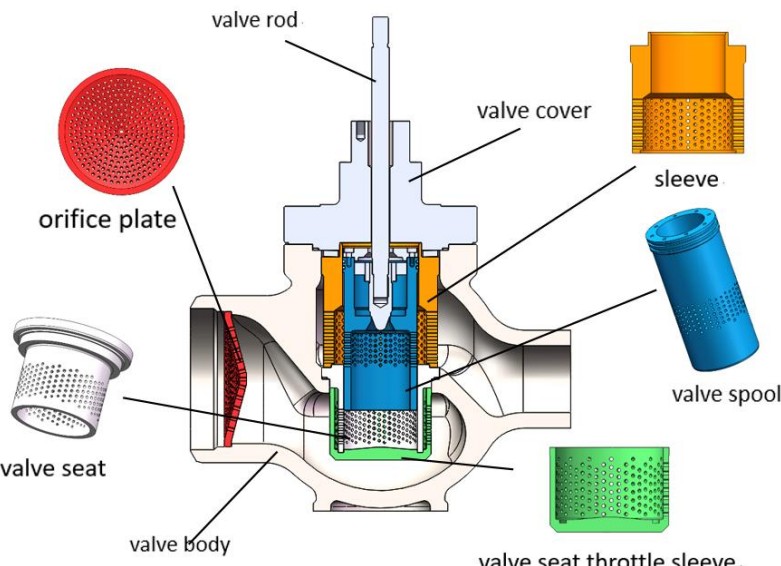

**Figure 1.** Three-dimensional model of the multi-stage sleeve-type control valve.

The valve consists of several components, including the valve body, valve bonnet, valve spool, valve seat (seat body and seat throttling sleeve), sleeve, stem, and throttling orifice plate. Various circular openings are incorporated into these components, with the diameter, number, and arrangement of the openings tailored according to the desired flow rate. Specifically, the sleeve orifices have a diameter of 6 mm and a length of 18.5 mm, with 300 through-holes arrayed in a circular pattern along the bottom circumference of the sleeve. The seat body orifices have a diameter of 5 mm and a length of 9.5 mm, with 300 through-holes arranged in a circular pattern along the bottom circumference of the seat body. The seat throttling sleeve orifices have a diameter of 5.5 mm and a length of 18.5 mm, with 315 through-holes arranged in a circular pattern along the bottom circumference of the seat throttling sleeve. The valve spool orifices have a diameter of 5 mm and a length of 12 mm, with 228 through-holes arranged in a circular pattern along the circumference. The throttling orifice plate contains 305 circular openings with a diameter of 6 mm and a length of 12 mm, distributed in a ring-shaped pattern.

The control valve regulates the flow by adjusting the position of the valve spool. The opening degree of the valve spool determines the flow area between the valve spool and the seat, thereby achieving the throttling of the steam.

## 3. Flow Field Calculation

### 3.1. Geometric Model and Meshing

We analyze the performance of a control valve under rated conditions. To streamline the computational process without compromising accuracy, a 3D model of the control valve is constructed using SolidWorks 2020. The extracted flow passages are appropriately simplified to facilitate numerical simulations. According to the standard IEC 60534-2-3 [21], the inlet and outlet effects of the flow passages significantly impact the steam flow within the control valve. To mitigate the influence of these effects on the results, the lengths of the inlet and outlet sections are set to 6 times and 10 times the diameter of the flow passage, respectively. The internal configuration of the control valve, referred to as the valve chamber, is depicted in Figure 2.

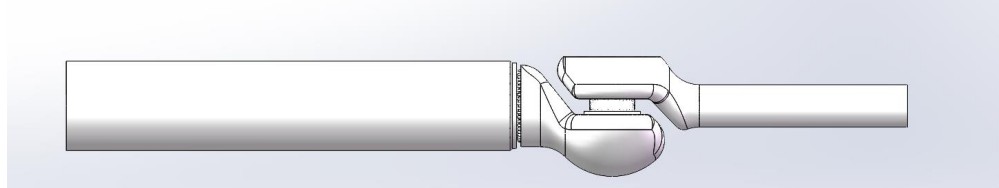

**Figure 2.** Control valve structure and internal flow path.

The mesh generation and discretization of the flow passage in a control valve are conducted using ANSYS ICEM CFD 2020. The three-dimensional model is appropriately simplified and modified to remove unnecessary edges and irrelevant regions that do not affect the computational results. The flow passage structure of the multi-stage sleeve control valve is relatively complex. To ensure grid independence in the calculations, mesh refinement is applied in regions with significant clearances between the valve seat and valve spool or areas with pronounced dimensional changes. Considering the numerous valve openings, finer local mesh refinement is implemented at locations such as the sleeve openings and the orifice plate openings to achieve a target y+ value of ≤30. Due to the irregularity of the extracted flow passage structure, structured grids are not suitable. To reduce the grid size, minimize memory usage, and enhance computational accuracy, a polyhedral meshing technique is employed for discretization.

Ensuring that the grid size is sufficiently fine to resolve wavelengths within the target frequency range is crucial for simulating the aerodynamic acoustic problems of valve systems. Therefore, an analysis and validation of control valve mesh density, wavelength magnitude, frequency range, and minimum grid size are performed. To determine the wavelengths within the target frequency range, the relationship between the speed of sound and frequency is utilized for wavelength calculations. A polyhedral meshing technique is employed in this study, providing a good grid resolution. The minimum grid size can be calculated based on the desired resolution, ensuring that the grid size is less than or equal to one-tenth of the wavelength, thereby capturing the highest and lowest frequency wavelengths.

Grid independence needs to be verified. The maximum face grid size is used as the independent variable, and the outlet mass flow rate is employed as the criterion to assess the reasonability of the grid partitioning. The verification scheme is presented in Table 1.

**Table 1.** Grid-independent verification.

| Maximum Size/mm | Number of Grids $\times$ $10^6$ | Mass Flow/(kg·s$^{-1}$) |
|---|---|---|
| 1.5 | 3.24 | 20.56 |
| 1.8 | 2.85 | 20.66 |
| 2.0 | 2.70 | 20.68 |
| 2.5 | 2.43 | 21.02 |
| 3.0 | 1.98 | 21.65 |
| 3.8 | 0.99 | 24.40 |

As shown in Table 1, among various partitioning schemes, the outlet flow rate remains relatively stable when the maximum face grid size is set to 2.0 mm. Therefore, a maximum face grid size of 2.0 mm is adopted for the mesh partitioning of the flow passage model. Multiple analyses and validations of grid independence are conducted to ensure that the influence of mesh density, wavelength magnitude, frequency range, and minimum grid size on the results is acceptable. The resulting discretized mesh of the flow passage model is depicted in Figure 3.

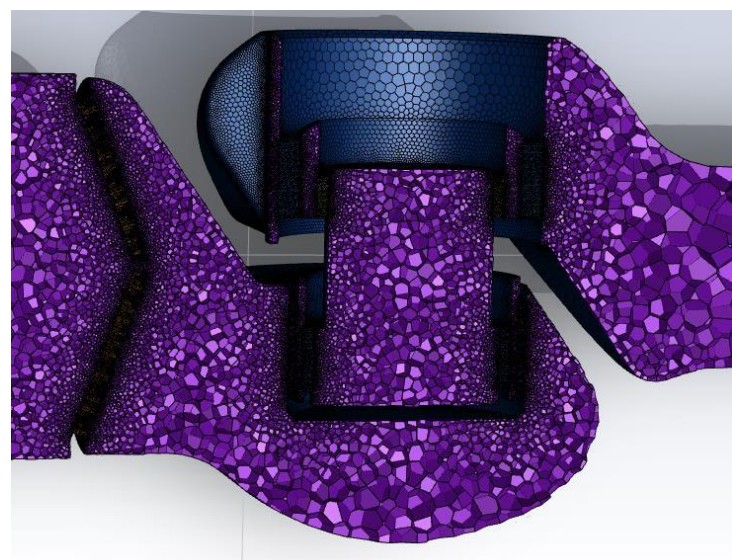

**Figure 3.** Three-dimensional grid diagram of flow channel model.

*3.2. Calculation Method*

The internal flow field of a control valve is numerically simulated using Fluent2020. A density-based solver is selected, with a steady-state formulation and the standard k-ε turbulence model. The boundary conditions used in the simulation are shown in Table 2. The medium is steam with a viscosity of $1.89 \times 10^{-5}$ kg/(m·s$^{-1}$) and a density of 35.7 kg/m$^3$. The operating pressure is 0 Pa, the boundary conditions are pressure inlet and pressure outlet, and the inlet temperature is 284.9 °C. The remaining surfaces of the flow passage are assumed to be adiabatic with a no-slip condition.

**Table 2.** Boundary condition settings.

| Medium | Vapor |
| --- | --- |
| Inlet pressure | 6.86 Mpa |
| Inlet temperature | 284.9 °C |
| Outlet pressure | 1 Mpa |
| Viscosity | $1.89 \times 10^{-5}$ kg/(m·s$^{-1}$) |
| Density | 35.7 kg/m$^3$ |
| Valve walls | Adiabatic no-slip boundary |

The solution algorithm is an implicit Roe-FDS method, and the flow, turbulent kinetic energy, and turbulent dissipation rate adopt a first-order upwind format. The solver control parameter (relaxation factor) controls the speed of convergence. Too large will lead to divergence at the initial calculation, and too small will lead to too small a convergence speed. Therefore, the size is set at the initial calculation, the number of steps is about 10,000, and the residual is set to increase when the residual is stable, as shown in Table 3. Finally, when the difference between import and export flow is less than 2%, it can be considered convergence.

**Table 3.** Residual setting.

| Residual Setting | Start of Calculation | After 10,000 Steps |
| --- | --- | --- |
| Courant number | 1 | 5 |
| Turbulent Kinetic Energy | 0.2 | 0.8 |
| Turbulent Dissipation Rate | 0.2 | 0.8 |
| Turbulent Viscosity | 0.2 | 1 |
| Solid | 0.2 | 1 |

According to the boundary conditions presented in Table 2, it is evident that there is a significant pressure difference at the inlet and outlet of the control valve. It caused the steam to flow through the valve in a highly turbulent state. Additionally, the throttling effect of components such as the valve spool, sleeve, and orifice plate further intensifies the turbulence of the steam. Therefore, in numerical simulations, it is essential to employ turbulence models alongside the fundamental fluid control equations.

The flow characteristics of the control valve are computed using the standard k-ε turbulence model. The standard k-ε model is a dual-equation turbulence model that gained wide application in engineering due to its good economy and high accuracy. The flow and noise problems of the regulating valve studied in this paper belong to the fields of fluid mechanics and aerodynamics, and the standard k-ε turbulence model is widely used in such large-scale or complex flow problems. It is a semi-empirical model based on the transport equations for turbulent kinetic energy and turbulent dissipation rate. The values of turbulent kinetic energy and turbulent dissipation rate are calculated using these two equations:

$$\frac{\partial}{\partial t}(\rho k) + \frac{\partial}{\partial x_i}(\rho k u_i) = \frac{\partial}{\partial x_j}\left[\left(u + \frac{u_t}{\sigma_k}\right)\frac{\partial k}{\partial x_j}\right] + G_k - \rho\varepsilon - Y_M \tag{1}$$

$$\frac{\partial}{\partial t}(\rho\varepsilon) + \frac{\partial}{\partial x_i}(\rho\varepsilon u_i) = \frac{\partial}{\partial x_j}\left[\left(u + \frac{u_t}{\sigma_\varepsilon}\right)\frac{\partial\varepsilon}{\partial x_j}\right] + C_{1\varepsilon}G_k\frac{\varepsilon}{k} - C_{2\varepsilon}\rho\frac{\varepsilon^2}{k} \tag{2}$$

where $G_k$ is the turbulent energy due to the mean velocity gradient, $Y_M$ is the expansion dissipation term, turbulent Prandtl number $\sigma_k = 1$, $\sigma_\varepsilon = 1$; $u_t$ is the turbulent viscosity; in addition, $C_u = 0.09$, $C_{1\varepsilon} = 1.44$, $C_{2\varepsilon} = 1.92$.

$$G_k = -\rho\overline{u_i'u_j'}\frac{\partial u_j}{\partial x_i} \tag{3}$$

$$Y_M = 2\rho\varepsilon\frac{k}{\gamma RT} \tag{4}$$

$$u_t = \rho C_u\frac{k^2}{\varepsilon} \tag{5}$$

*3.3. Calculation Results and Analysis*

The Mach number, turbulent kinetic energy and static pressure distribution of the regulator under the rated working conditions were obtained by numerical simulation method, and the flow performance of the regulator was analyzed. The Mach number is shown in Figure 4a.

From Figure 4a, it can be observed that as the steam medium passes through the sleeve, the flow velocity increases due to the decrease in the flow cross-sectional area. The flow velocity is higher in the orifice plate as compared to the sleeve due to the smaller flow area in the valve spool orifice. After converging in the middle of the sleeve, the steam flows towards the valve body and the seat throttle sleeve through the flow passage, with the velocity increasing sequentially through the two layers of the sleeve. As the steam flows out of the seat throttle sleeve and enters the valve body, a faster flow velocity can be observed in the region, with a smaller flow area in the lower chamber of the valve body. At the exit of the throttle orifice plate, supersonic flow is observed, with a maximum Mach number reaching 1.28.

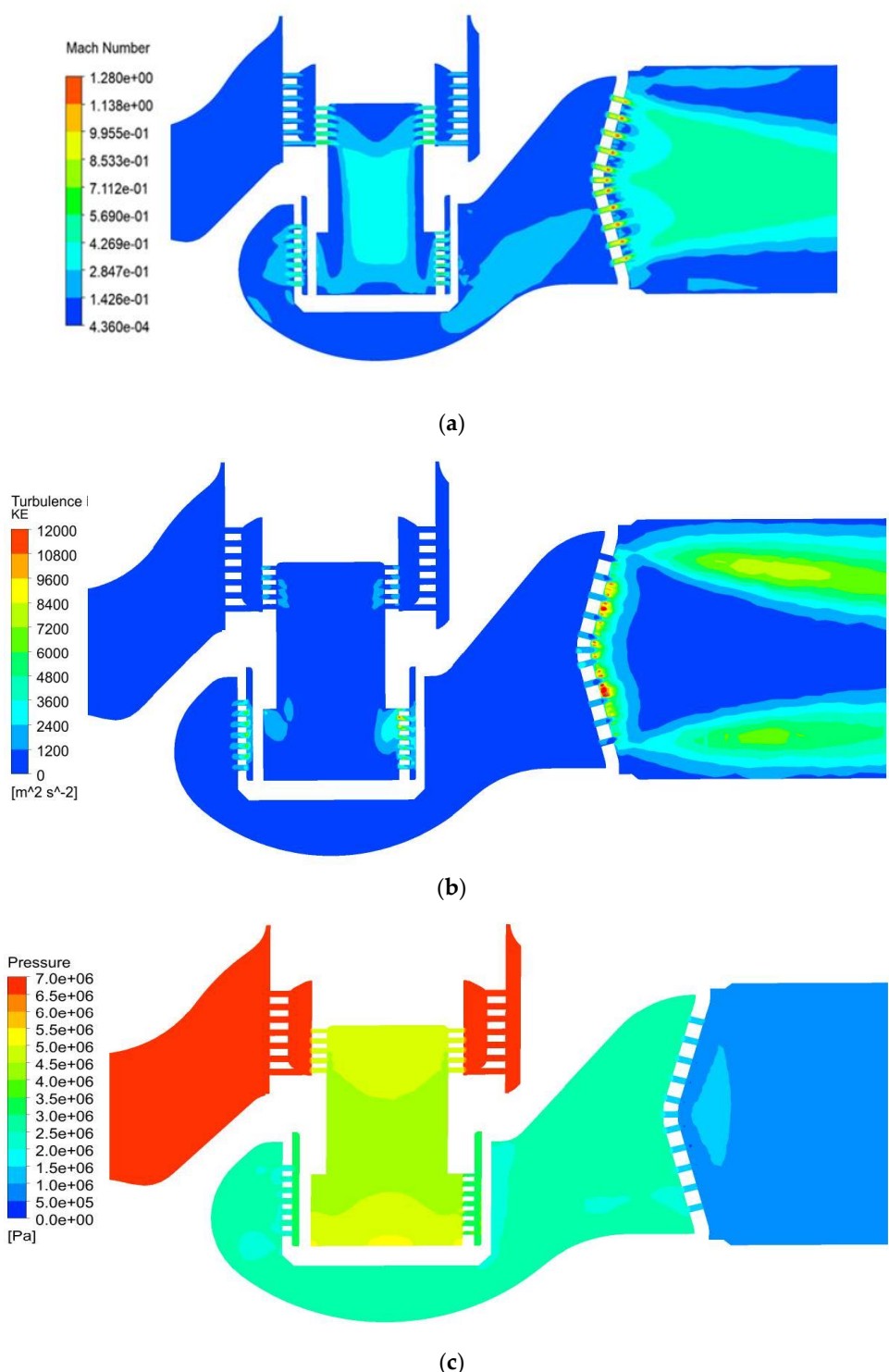

**Figure 4.** (**a**) Cross-sectional velocity cloud. (**b**) Cross-sectional turbulent kinetic energy cloud. (**c**) Cross-sectional pressure cloud.

Figure 4b shows the contour plot of turbulent kinetic energy in the control valve cross-section. The essence of turbulence is the irregular motion of a fluid when it flows at too high a velocity. To quantify the intensity of turbulence, known as turbulence intensity, occurring in the flow through the control valve, the analysis of turbulent kinetic energy is required. As shown in Figure 4b, the maximum turbulent kinetic energy occurs at the exit of the throttle orifice plate, measuring approximately $1.2 \times 10^4 \ \text{m}^2/\text{s}^2$.

Figure 4c presents the contour plot of pressure in the control valve cross-section. The inlet and outlet pressures of the control valve are constant. From the static pressure contour plot in Figure 4c, it can be observed that the pressure drop is not significant as the steam flows through the first layer of the sleeve. However, a noticeable pressure drop occurs as the steam passes through the second layer of the sleeve. A similar pressure drop pattern is observed in the seat sleeve. This indicates that the sleeve structure plays a role in pressure reduction. The pressure distribution in the lower chamber of the valve body is relatively uniform, and after passing through the throttle orifice plate, the pressure significantly decreases. The greatest pressure drop occurs in the upper and lower regions of the throttle orifice plate.

## 4. Sound Field Calculation

### 4.1. The FW-H Acoustic Fitting Theory

Calculation of the acoustic field requires the extraction of the acoustic source term using the FW-H acoustic wave equation. The FW-H equation is derived from the Navier–Stokes equations in fluid mechanics. The Lighthill equation is then extended to incorporate solid boundaries, and the influence of moving objects on the acoustic field is included by Ffowcs Williams and Hawkings et al. [22,23].

In this study, the integration domain for solving the FW-H equation refers to the region inside the control valve where the steam medium flows. The entire flow domain is discretized and divided into finite computational cells, and the flow equations are solved on each computational cell. This allows for the discrete solution of the flow field on a global scale.

The magnitude of the aerodynamic noise at a given location can be calculated by the area fraction in the FW-H acoustic model, and the FW-H equation is shown below:

$$
\begin{aligned}
\left( \frac{\partial^2}{c^2 \partial t^2} - \frac{\partial^2}{\partial x_i^2} \right) p'(x_i, t) = {} & \frac{\partial}{\partial_t} \{ [\rho_0 v_n + \rho(\mu_n - v_n)] \delta(f) \} - \\
& \frac{\partial}{\partial x_i} \left\{ \left[ -P'_{ij} n_j + \rho \mu_i (\mu_n - v_n) \right] \delta(f) \right\} + \frac{\partial^2}{\partial x_i \partial x_j} [T_{ij} H(f)]
\end{aligned}
\tag{6}
$$

where the left side of the equation is the fluctuation operator, and the sound pressure level at the observation point at moment t, respectively; the right side of the equation indicates the monopole source, dipole source, and quadrupole source, respectively [12]. $f$ is the boundary control surface function of the moving object, $T_{ij}$ is the Lighthill tensor; $\rho$, $\mu_i$ and $P$ are the density, velocity, and stress tensor, respectively; $\delta_{ij}$ is the Kronecker function; the Heavi-side function, $\delta(f)$ is the Dirac function; $x$ is the spatial coordinate, and the indicators $i$ and $j$ denote the coordinate axis direction components.

### 4.2. Calculation Method

The computational fluid dynamics (CFD) method is employed to accurately calculate the acoustic field inside the control valve. The steady-state flow field calculation results are used as the initial conditions, and the transient flow field analysis of the control valve is performed using the k-ξ method to calculate the flow-induced noise. First, select the option Transient in the model tree node General to start transient calculations. Select Ffowcs-Williams & Hawkings in the node Models and check the options "Export Acoustic Source Data in ASD Format" and "Compute Acoustic Signals Simultaneously". The acoustic model setup is shown in Figure 5.

Finally, click Define Sources to pop up the sound Source definition dialog box, select Source Zones as the wall, set Source Data Root File Name, and set Write Frequency. Set the Number of Time Steps per File. The acoustic source setup is shown in Figure 6.

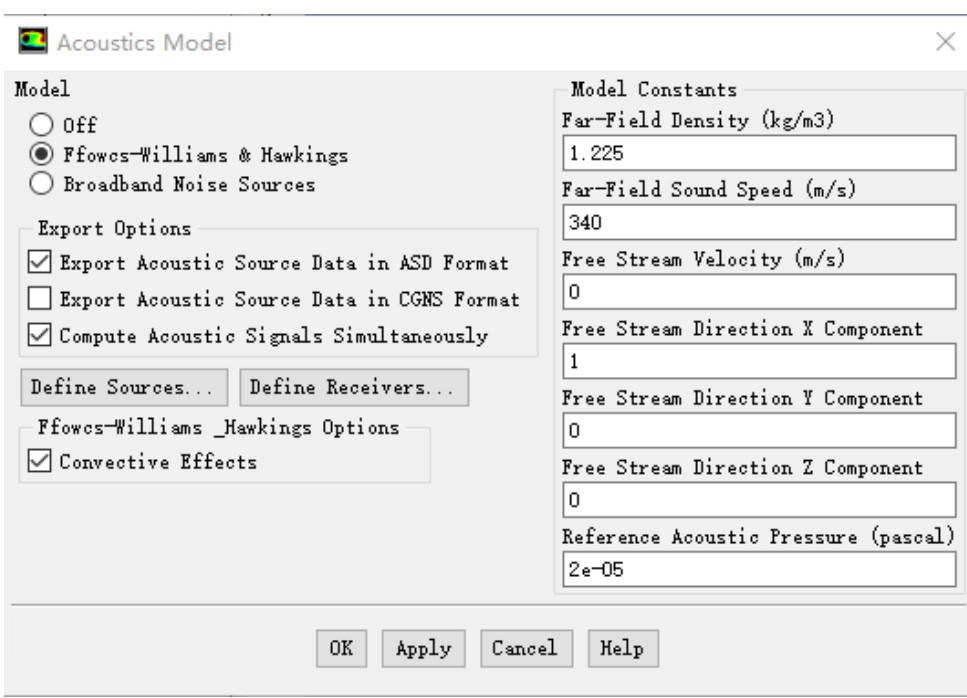

**Figure 5.** Acoustic model setup.

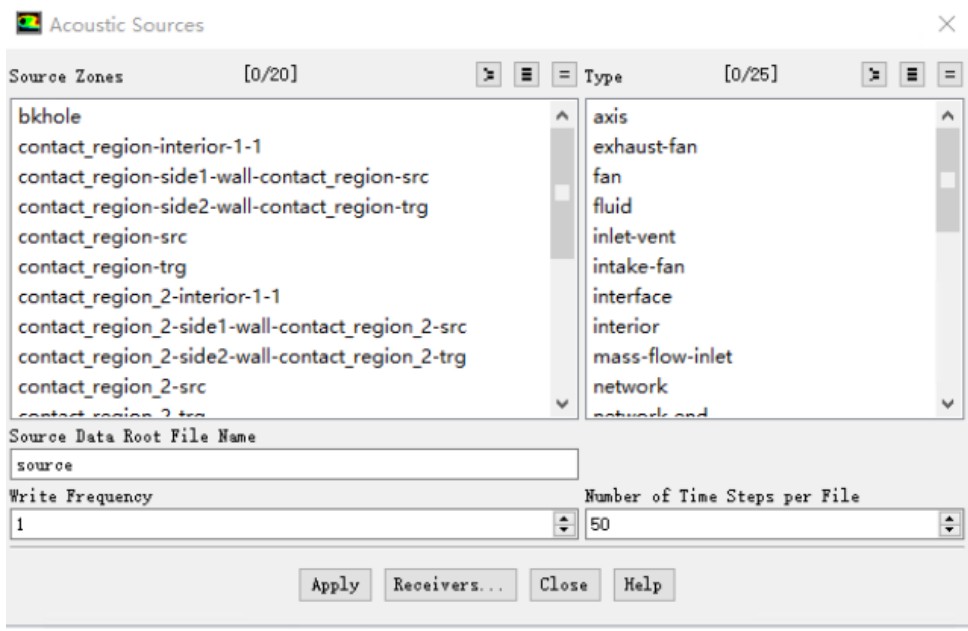

**Figure 6.** Acoustic sources setup.

Based on experience, the noise generated by the valve is mainly concentrated within the 0–10,000 Hz range. Therefore, the time step for transient calculation is chosen as $5 \times 10^{-5}$ s, and the total number of time steps is set to 2000 steps. According to the spectral characteristics of flow noise, the sound pressure values at various points gradually decrease as the frequency increases and eventually reach a steady-state value. After the calculation is completed, process the outputted acoustic files in the Acoustic Signals section. After the calculation, the output Acoustic file is processed in Acoustic Signals. Using FFT Plots in the Results node, the relationship between sound pressure level and frequency of noise monitoring points is obtained. The relationship between sound pressure level and frequency multiplication and weighted sound pressure level can also be obtained.

### 4.3. Calculation Results and Analysis

Figure 7a displays the contour plot of sound power level distribution in the steam passage inside the valve body. From the data in Figure 7a, it can be observed that the sound power level is lowest in the inlet pipe. The sound power level is relatively high in the throttle sleeve within the throttle seat area and its surroundings. The maximum sound power level is located behind the throttle orifice plate, suggesting that the main source of noise is situated behind the orifice plate.

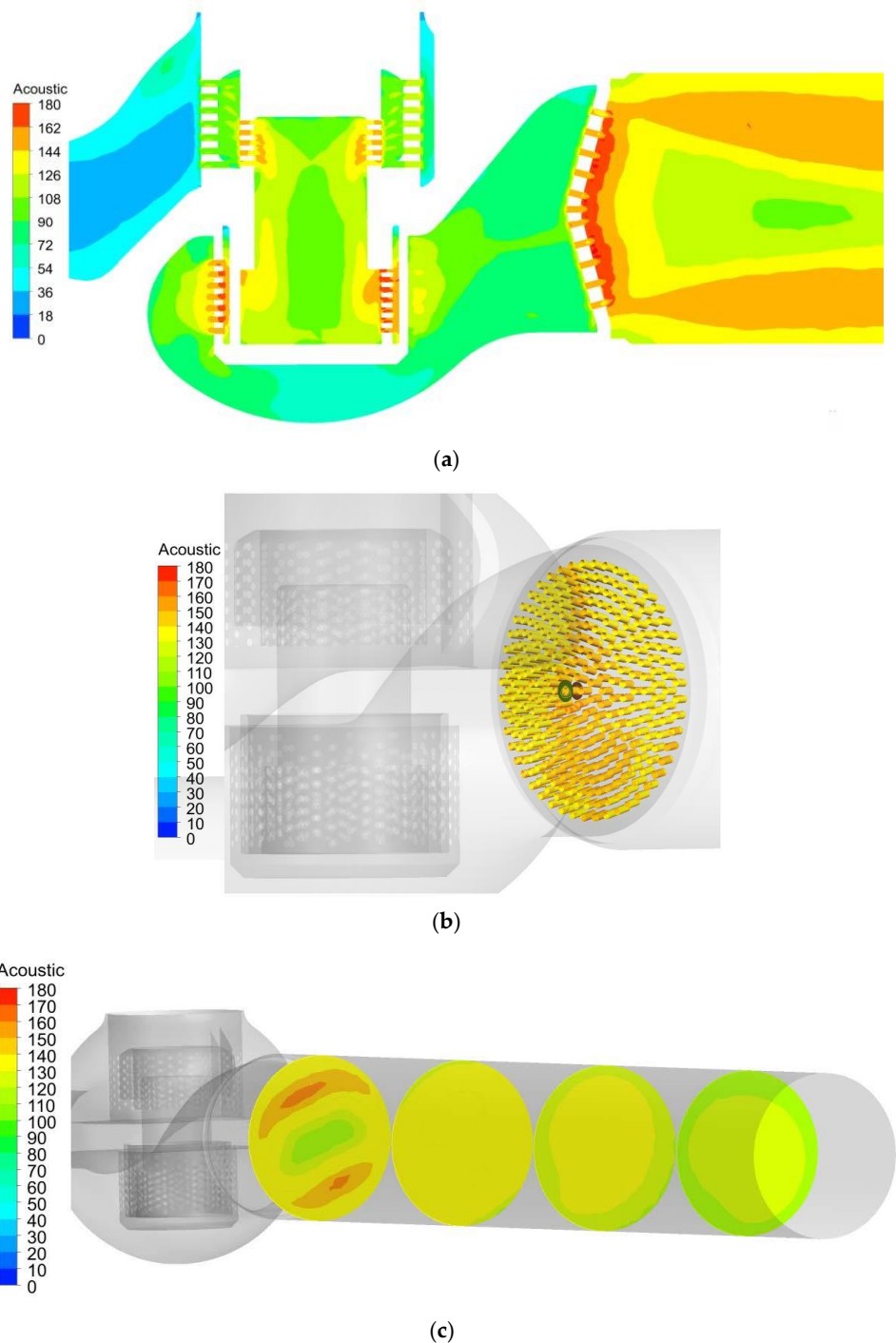

(**a**)

(**b**)

(**c**)

**Figure 7.** (**a**) Cloud diagram of sound power level distribution of control valve flow channel. (**b**) Cloud of sound power level distribution on the surface of small holes of the orifice plate. (**c**) The average sound power level of different sections of the outlet flow channel.

Based on the above analysis, it is evident that the orifice plate is the main source of noise in the valve. Therefore, it is necessary to analyze the sound power level on the surface of the orifice plate. The distribution of sound power level on the orifice plate surface is related to the shearing force exerted by the steam on the orifice plate, as shown in Figure 7b. Since the steam velocity is high when passing through the small orifice, the shearing force on the orifice is high, resulting in a higher sound power level on the surface. The sound power level on the orifice plate surface ranges from 140 dB to 180 dB. Thus, the entire orifice plate exhibits high sound power levels and represents a significant source of aerodynamic noise.

Figure 7c presents the contour plot of average sound power levels at different sections of the control valve outlet passage. It indicates a gradual decrease in the average sound power level as the steam flows towards the outlet. Overall, there is a circular distribution pattern at different sections, with the lowest sound pressure level in the central region and increasing towards the outer regions. However, at sections far from the outlet, the sound power level decreases near the pipe walls.

According to the latest standard "GB/T 17213.15-2017 Industrial Process Control Valves Part 8-3: Reflections on the Prediction Method for Aerodynamic Flow Control Valve Noise" [24], monitoring of the noise level around the valve is necessary.

The total sound pressure value of aerodynamic noise reflects the intensity of noise at different positions in the steam control valve acoustic field. To study the directivity of aerodynamic noise in the acoustic field, it is necessary to strategically place monitoring points. The monitoring points are set by taking a point 1 m downstream of the valve as the center and creating two concentric circles with radii of 1 m and 2 m, each containing 12 evenly spaced monitoring points. A total of 25 monitoring points are set in the acoustic field, including the center point. The distribution and numbering of monitoring points are shown in Figure 8.

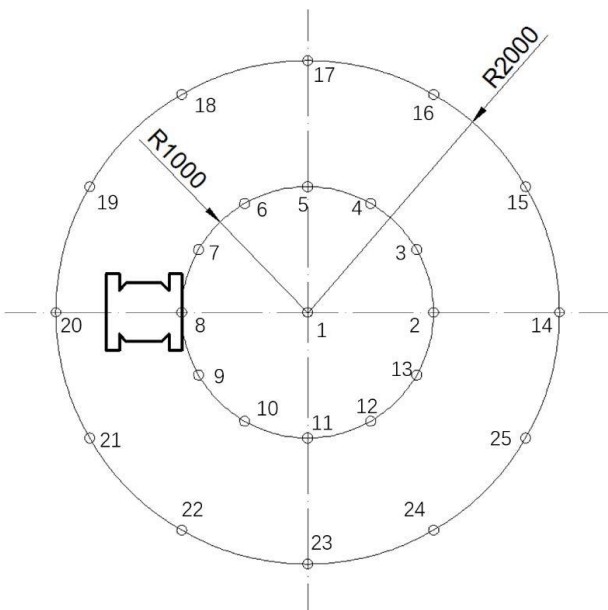

**Figure 8.** Noise monitoring point settings.

The total sound pressure level is recorded at each monitoring point to analyze the directivity of aerodynamic noise in the steam control valve. Figure 9 shows the distribution curve of total sound pressure levels along the circular monitoring points. From Figure 7, it can be observed that the distribution of monitoring points roughly forms two elliptical shapes. Additionally, the directivity of aerodynamic noise follows a similar pattern. In the acoustic field, the distribution of total sound pressure levels along the circular path with R = 1000 exhibits a "double-ended pointer" shape, with significantly higher sound

pressure levels near the upstream and downstream ends compared to other directions. The distribution of total sound pressure levels along the circular path with R = 2000 is similar to that of R = 1000. This analysis indicates that the aerodynamic noise of the steam control valve exhibits noticeable directivity in the acoustic field, with higher noise levels in the regions near the upstream and downstream.

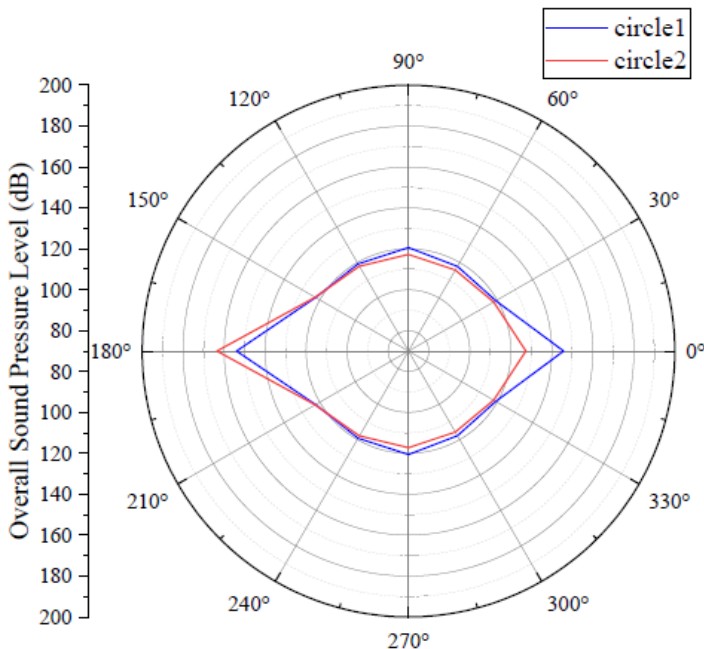

**Figure 9.** Total sound pressure level distribution curve on the circumference of the monitoring point.

From the monitoring points and the distribution curve of total sound pressure levels, it is evident that the aerodynamic noise is strongest at monitoring point 8. Therefore, Figure 10 analyzes the 1/3-octave band curve at monitoring point 8. The sound pressure level at monitoring point 8 reaches 145 dB at low frequencies and stabilizes at around 130 dB at high frequencies.

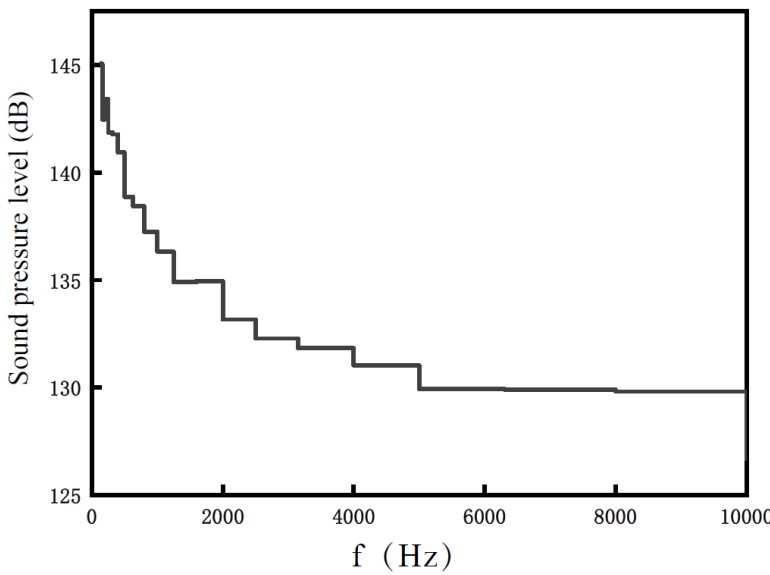

**Figure 10.** 1/3 octave plot under valve monitoring point 8.

### 4.4. Numerical Simulation Method Validation

According to the latest standard «GB/T 17213.15-2017 Industrial process control valves Part 8-3 noise considerations: methods for predicting the noise generated by aerodynamic flow through control valve» [24] theoretical calculation of pneumatic noise monitoring points. The calculation method is as follows:

The sound pressure level inside the valve on the pipe wall is calculated as follows:

$$L_{pi} = 10\lg\left[\frac{(3.2 \times 10^9)\,W_a\rho_2 c_2}{D_i^2}\right] + L_g \tag{7}$$

Propagation loss calculation formula:

$$TL(f_i) = 10\lg(A_1 B_1) - \Delta TL \tag{8}$$

$$A_1 = \left(8.25 \times 10^{-7}\right)\left(\frac{c_2}{t_s f_i}\right)^2$$

$$B_1 = \frac{G_x(f_i)}{\frac{[\rho_2 c_2 + 2\pi t_s f_i \rho_s \eta_s(f_i)]}{415 G_y(f_i)} + 1}\left(\frac{p_a}{p_s}\right)$$

External sound pressure level at 1 m outside the pipe wall:

$$L_{pAe,1m} = L_{pi}(f_i) + TL(f_i) - 10\lg\left(\frac{D_i + 2t_s + 2}{D_i + 2t_s}\right) \tag{9}$$

Weighted sound pressure level at 1 m outside:

$$L_{pAe,1m} = 10\lg\left(\Sigma_{i=1}^{N=33}10^{\frac{L_{pAe,1m}(f_i) + \Delta L_A(f_i)}{10}}\right) \tag{10}$$

where $W_a$ is the sound power downstream of the valve; $c_2$ is the downstream speed of sound; $\rho_2$ is the fluid density; $\eta_s(f)$ is the structural dissipation coefficient; $G_x, G_y$ are both frequency coefficients; $D_i$ is the internal diameter of the downstream pipe; $t_s$ is the pipe wall thickness; $f_i$ is the 1/3 octave frequency; $L_{pi}(f_i)$ is the internal sound pressure level at frequency $f_i$; $TL(f_i)$ is the propagation loss at frequency $f_i$; and $\Delta L_A(f_i)$ is the A-weighting factor at frequency $f_i$.

For valves using high-temperature steam as the medium, the main source of noise is aerodynamic noise, while flow-induced vibration noise accounts for a relatively small proportion. Therefore, qualitative analysis of aerodynamic noise in the valve can assess the magnitude of noise at the valve root and the effectiveness of noise reduction. In conclusion, the comparison and validation of aerodynamic noise numerical simulations can be conducted using theoretical formulas and Fluent computations.

Table 4 presents the required parameters and relevant calculations for the theoretical calculation of aerodynamic noise in the valve, using the boundary condition data as an example.

**Table 4.** Valve theoretical calculation parameters.

| Type Fluid | Vapor |
| --- | --- |
| Mass flow rate | m = 20 kg/s |
| Valve inlet absolute pressure | $P_1$ = 6.86 MPa |
| Valve outlet absolute pressure | $P_2$ = 1 MPa |
| Inlet density | $\rho_1$ = 35.7 kg/m$^3$ |

**Table 4.** *Cont.*

| Type Fluid | Vapor |
|---|---|
| Inlet absolute temperature | $T_1$ = 558.05 K (284.9 °C) |
| Specific heat ratio | $\gamma$ = 1.885 |
| Molecular mass | M = 19.8 kg/kmol |
| Required Cv | Cv = 101 |
| Valve size | DN125 |
| Valve outlet diameter | D = 0.241 m |
| Internal pipe diameter | $D_1$ = 0.241 m |
| Differential pressure ratio | x = 0.985 |
| Absolute Vena contracta pressure at subsonic flow conditions | Pvc = −1,482,098 pa |
| Vena contracta differential pressure ratio at critical flow conditions | $x_{vcc}$ = 0.542 |
| Differential pressure ratio at critical flow conditions | $X_c$ = 0.438 |
| Recovery correction factor | $\alpha$ = 0.817 |
| Differential pressure ratio at breakpoint | $X_B$ = 0.683 |
| Differential pressure ratio where region of constant acoustical efficiency begins | $X_{CE}$ = 0.944 |
| Regime definition | $X_{CE}$ < X, Regime V. |
| Hydraulic diameter of a single flow passage | $d_H$ = 0.575 |
| Diameter of a circular orifice | $d_o$ = 3.582 |
| Valve style modifier | $F_d$ = 0.16 |
| Jet diameter | $D_j$ = 0.007 |
| Strouhal number of free jet at peak frequency | $S_{tp}$ = 0.2 |
| Correction factor of sound effect coefficient | $A_\eta$ = −3.8 |

The medium of the multi-stage sleeve control valve used in this paper was steam. When the noise is predicted by the IEC 60534-8-3 standard theory, the theoretical noise calculation cannot take into account the noise caused by the reflection of the outer surface of the valve body and the internal pipe assembly, mechanical vibration, medium flow state, and other factors. When numerical simulation is used to calculate noise, the above factors will affect the numerical simulation results. Therefore, there are some differences between the noise pressure level predicted by theory and the noise pressure level calculated by numerical simulation. Figure 11 shows a bar chart comparing the simulated and theoretical data for noise at selected monitoring points 2–8.

As depicted in Figure 11, we found that among the selected monitoring points, the noise of the theoretical calculation and the numerical analysis of the monitoring point 8 was 157.86 dB (A) and 161.95 dB (A), respectively, and the difference between the calculation results of the two was 4.09 dB (A). The noise values of the other monitoring points were between 110 and135 dB (A), and the difference between the calculated results was the smallest at the monitoring point 6, which was 1.16 dB (A). The error of the monitoring points was less than 5%; this method verifies the validity of the numerical simulation method. Using the valve design optimization, the numerical simulation method was used to verify the valve noise size problem, so that it was within the allowable range of industrial noise, thus reducing the engineering cost.

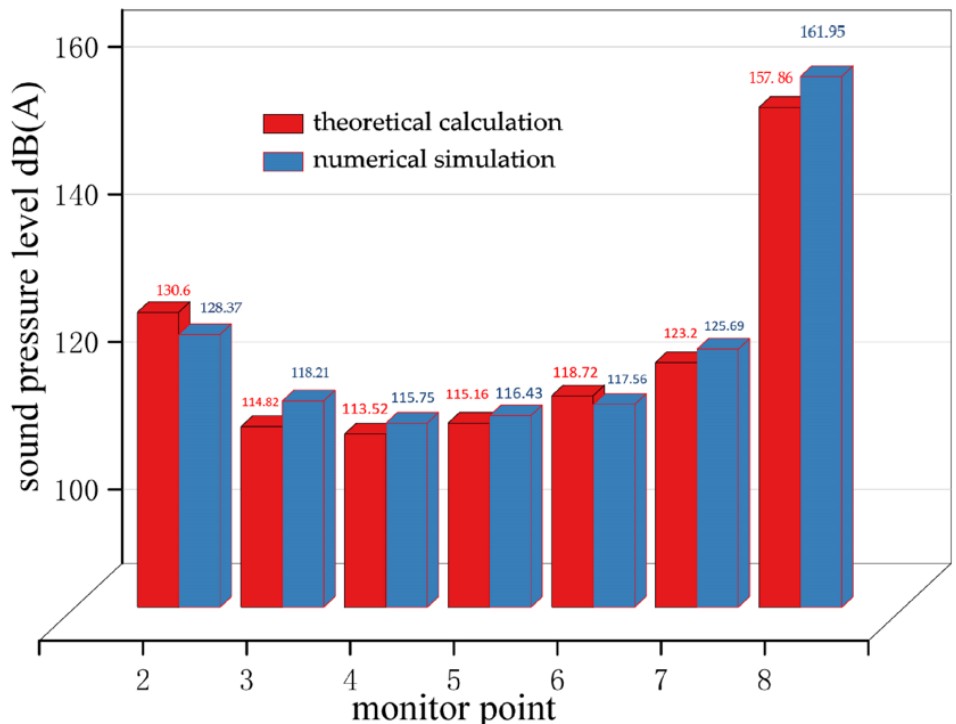

**Figure 11.** Comparison of numerical simulation and theoretical data of some monitoring points.

## 5. Conclusions

In the present investigation, the flow and sound fields of the steam medium in a multi-stage sleeve-type control valve were examined under rated operating conditions. The cause and distribution of noise were revealed. The following conclusions were drawn:

1.  Analysis of the steam medium within the control valve under operating conditions revealed important aspects such as the Mach number, turbulence, and static pressure distribution. It was observed that the disturbance caused by high-speed flow in the valve components increased the intensity of turbulence energy. Consequently, the high turbulence energy led to steam pressure pulsations, which served as a primary source of noise.

2.  The study revealed a symmetrical distribution of sound pressure levels along the tube and valve system of the control valve. Furthermore, it was observed that pneumatic noise exhibited directional characteristics, with significantly higher noise levels detected upstream and downstream of the valve compared to other regions.

3.  Among the many monitoring points selected, the noise of monitoring point 8 was 157.86 dB (A) in theoretical calculation and 161.95 dB (A) in numerical analysis, and the difference between the two results was 4.09 dB (A). At monitoring point 6, the difference was 1.16 dB (A). The error of the monitoring points was less than 5%, which verifies the effectiveness of the numerical simulation method.

Overall, this study provided valuable insights into the flow and sound characteristics of steam in a multi-stage sleeve-type control valve, highlighting the significance of turbulence energy and the directional nature of pneumatic noise. The validated numerical simulation approach offers a reliable tool for optimizing valve noise and enhancing engineering practices.

**Author Contributions:** Methodology, J.J. and B.Z.; validation, J.J.; formal analysis, Y.S. and B.Z.; resources, X.M. and Y.S.; data curation, J.J.; writing—original draft preparation, J.J.; writing—review and editing, J.J. and X.M.; supervision, X.M. and D.L.; project administration, Y.S. and B.Z. All authors have read and agreed to the published version of the manuscript.

**Funding:** This research received no external funding.

**Data Availability Statement:** The data that support the findings of this study are available from the corresponding author.

**Conflicts of Interest:** The authors declare no conflict of interest.

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
