# Peer review of "Pneumatic Noise Study of Multi-Stage Sleeve Control Valve"

_processes, doi:10.3390/pr11092544_

Round 1
Reviewer 1 Report
The overall findings from the study is good but not up to the highest level. Proofreading is required along with a solid English writing base.
The English need to be improved. Sometimes long sentences used in the manuscript are not making any sense. Poor sentence structure.
Author Response
Dear Editor and Reviewers,
Thank you very much for your recognition of our manuscript. The authors thank editors and reviewers for their comments to improve the quality of the paper. Your comments will be of great help to our future paper writing. We have carefully studied these comments and have made corrections. The modified content has been marked in the main body. The title of the article is " Pneumatic Noise Study of Multi-Stage Sleeve Control Valve." (Manuscript ID: processes- 2435806) Looking forward to your approval.
Considering the Reviewer’s suggestion. We have found professional English writers to revise the article. We have carefully revised the articles to ensure the quality of their content and the level of writing. Thanks for your suggestion.
Reviewer 2 Report
1. The quality of Figure 1 is poor and dark to distinguish the components of valve. It must be updated.
2. How much is yplus value? Also, for simulate the aeroacoustics from valve system, the grid size must enough to resolve the wavelength of target frequency range. It would be described in this paper.
3. How did you make grids around numerous number of holes?Did you make the grids on that or use any porous media scheme?
4. The version of fluent or commercial programs should be described.
5. Why did you use the standard k-epsilon turbulence model?
6. The contours don't have enough resolution and information to support the current description.
7. When the FW-H eqns. were solved, what was the integral surface or zone?
8. After section 4.2, the results have to be described with frequency and also, the results aren't enough to support current investigation.
Must be improved
Reviewer 3 Report
The work performed is interesting, but the paper lacks important information. I recommend reconsidering the publication of the paper until the authors incorporate these crucial additions.

The article's English needs improvement. We highly recommend consulting with a native English speaker for assistance.
Round 2
Reviewer 1 Report
All the previous concerns are properly addressed. Satisfied with the overall improvements of the paper.
Author Response
Thanks very much for your attention to our paper.
Very sincerely yours,
Jia Jianbo
Reviewer 2 Report
The overall quality of the current paper has improved, but it could be better.
The expression of numerical methods would only be improved with technical words used in commercial programs.
Reviewer 3 Report
Thank you for your answers. Everything is clear now. I would recommend your paper for publication.
Author Response

(The authors gave the same response as above.)

Round 3
Reviewer 2 Report
-
-